# On Detecting COVID-Risky Behavior from Smartphones

Thomas Hartvigsen*
tomh@mit.edu
Massachusetts Institute of Technology

Walter Gerych*
wgerych@wpi.edu
Worcester Polytechnic Institute

Marzyeh Ghassemi
mghassem@mit.edu
Massachusetts Institute of Technology

## ABSTRACT

Detecting risky behavior using smartphones and other mobile devices may help mitigate the spread of infectious diseases. However, the privacy concerns introduced by individualized activity recognition may counteract potential benefits. As an example, consider a public health official gauging their message's efficacy. Machine learning and data mining methods may help them understand how their local population's behavior changes, but aggressive surveillance can severely hinder individual privacy and cause disproportionate harm to disadvantaged groups. In this work, we benchmark a series of machine learning algorithms predicting high-risk behaviors—going to bars and gyms, attending parties, and riding on buses—given only low-level smartphone sensor data. We find that models trained to perform these challenging tasks are largely unreliable and should be avoided in practice, though their predictions are significantly better than random.

## KEYWORDS

public health, digital health, time series

**ACM Reference Format:**
Thomas Hartvigsen, Walter Gerych, and Marzyeh Ghassemi. 2022. On Detecting COVID-Risky Behavior from Smartphones. In *epiDAMIK 2022: 5th epiDAMIK ACM SIGKDD International Workshop on Epidemiology meets Data Mining and Knowledge Discovery, August 15, 2022, Washington, DC, USA*. ACM, Washington, D.C., USA, 4 pages.

## 1 INTRODUCTION

Risky behavior during a pandemic can create outbreaks. To combat this, officials release public health messaging to alert to educate the population [7]. But how can we quickly and automatically tell when a population's behavior aligns with public health messaging? One promising direction may be through Human Activity Recognition via mobile devices [12], like smartphones or watches, which contain sensors that rapidly measure movement. By detecting what activities people are performing and where, public health may be improved, particularly during a pandemic where success requires community-level cooperation. Accurately identifying people's behavior using data collected from wearable devices is becoming feasible [21], and we believe that this approach may play a role in mitigating the spread of future diseases as well as continuing to mitigate COVID-19.

*Equal Contribution

Consider, for example, early COVID-19 spread in the US. Public health officials knew that drinking at bars puts people at high-risk of contracting and spreading COVID-19, a message they began broadcasting to their communities. However, public health has a notoriously-slow feedback loop: determining a messaging strategy's efficacy may take so long that the system has completely changed, leading to public mistrust [24]. This was indeed the case in the US, which has hosted one of the earliest, largest, and longest outbreaks of COVID-19. Ground-level monitoring of population behavior via data mining could be a promising approach to expedite the feedback loop for public health officials.

However, recognizing a population's activities at scale means using individuals' data. This can cause harmful invasions of privacy [8]. In particular, individual surveillance disproportionately harms historically-disadvantaged communities [9]. These negatives should discourage the use of mobile data to detect behavior for public health.

On the technical side, detecting and predicting human behavior from smartphones and other wearable devices is a burgeoning field, with many successful applications, particularly for COVID [14, 16, 18]. Still, while success at detecting human behaviors is increasing, significant challenges remain. Wearable sensor time series are often long, noisy, and high-dimensional [1, 15]. Self-reported human behavior labels are frequently sparse, missing, and temporally biased [10, 11, 17, 27]. Still, success at detecting human behaviors is increasing.

We investigate the feasibility of Human Activity Recognition for detecting COVID-risky behaviors by presenting experimental results on the publicly-available EXTRASENSORY smartphone sensor dataset [26]. This dataset contains smartphone sensor data collected in-the-wild: people went about their daily lives and annotated what actions they were performing. Four of the reported activities pose significant risks of spreading COVID-19, especially early during the COVID-19 pandemic when personal protective equipment was sparse: *Going to a Bar*, *Going to the Gym*, *Going to a Party*, and *Riding on the Bus*. We train a range of machine learning models to predict these labels and compare a variety of domain-appropriate data preprocessing techniques.

Overall, we find that, surprisingly, some of the COVID-risky behaviors are somewhat predictable: by upsampling sparse labels or downsampling common labels, we find balanced accuracies of over 0.7 for all tasks, with some even breaching 0.8, *without* modifying testing data at all. As expected, XGBoost [5] overall performs better than Logistic Regression, a Support Vector Machine, and a Multilayer Perceptron for this dataset. These accuracies are higher than expected, indicating that some organizations may investigate similar approaches in the near future. While we advocate against their

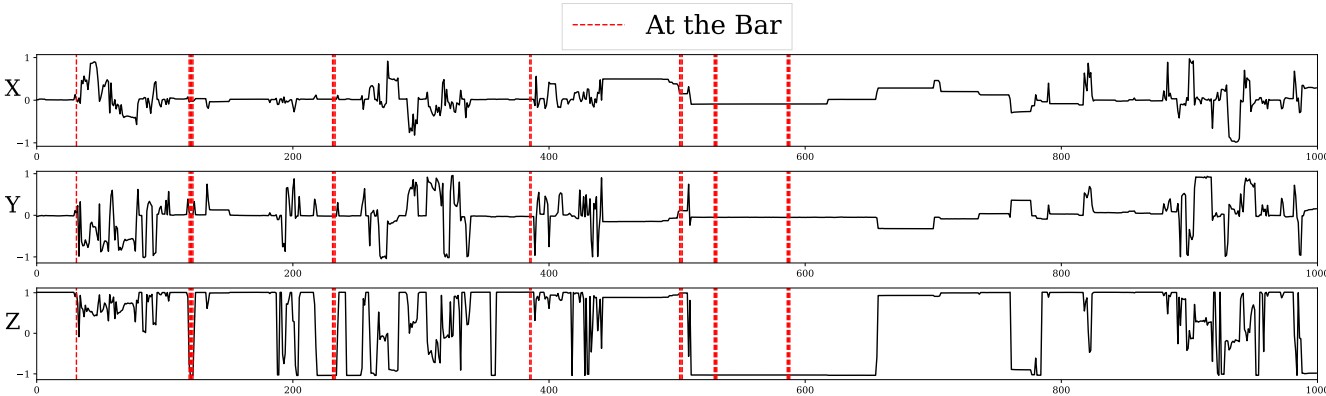

**Figure 1: Example of accelerometer data from ExtraSensory. Rows are the X, Y, and Z from the smartphone accelerometer, respectively while vertical red lines indicate when this user reported they were at a bar.**

immediate deployment, legislation surrounding these technologies must be prioritized now in order to prevent the harms of privacy breaches, which particularly harm historically-disadvantaged groups.

## 2  RELATED WORKS

Prior work has shown that smartphone sensor data is predictive of not only a person's physical location, but also the semantic meaning of the location they are inhabiting. These *semantic locations* include human-meaningful places like someone's home, their place of work, or dining establishments [23]. This is of particular interest to our work, as some semantic locations like being on a crowded bus or in a busy bar are risky behaviors during pandemics. While early works mainly used GPS data [4, 13], more recent works have shown that other signals such as accelerometer and gyroscope readings are predictive of semantic location [19, 23]. This indicates that *dynamic* semantic locations that do not always exist in the same location, such as *parties*, might be identifiable using smartphone data.

Using smartphone sensed data to classify the behaviors of individuals is also an active area of research [22, 26]. Prior work has focused on using smartphone sensor data to detect atomic activities such as *walking* or *sitting* [2], complex activities such as *having dinner* [25], and contexts which consist of sets of activities and states such as *walking in public* [3, 26]. Since risky behaviors such as drinking in a crowded bar consist of a sequence of activities and contexts, the predictive power of smartphone data for such tasks indicates that risky behavior can likewise be detected.

## 3  DATASET

We experiment with the publicly-available ExtraSensory [26] human activity recognition dataset, which contains smartphone sensor data collected across one week while 60 participants labeled which actions they performed and when. The mobile application responsible for data collection continuously logged their movement via sensors (e.g., accelerometer and gyroscope). These data were collected in the wild, so participants were left to their own devices for the duration of the study with no prescribed behavior. This means that some people self-annotated behaviors that we deem

| Class | Number of Positive | Number of Negative |
|---|---|---|
| At a Bar | 63 | 3257 |
| At the Gym | 132 | 4303 |
| At a Party | 152 | 4487 |
| On a Bus | 280 | 15249 |

**Table 1: Number of positives and negatives for each class.**

risky for COVID-19: *Going to a Bar*, *Going to the Gym*, *Going to a Party*, and *Riding on the Bus*. Compared to the rest of the dataset, these labels are quite rare, leading to large imbalance.

We extract 6 key variables from two sensors in ExtraSensory: The X, Y, and Z directions of the accelerometer, and X, Y, and Z directions of the gyroscope. Using only these features demonstrates it can be possible to mine complex patterns from only raw sensor measurements. We then extract windows of 10 timesteps and for each label independently, assign windows to be positive when any of the 10 timesteps was labeled positively by a participant. Experimentally, we ranged the window size from 5 to 20 and found the results to be robust across sizes. Naturally, the labels we've chosen as COVID-risky are rare compared some other classes in ExtraSensory, like walking or running, and so these data are highly imbalanced. To preserve realistic test cases, we leave all testing data untouched in the bulk of our experiments. Further, since there is little overlap between windows from different labels, we extract *one unique dataset per label* from the full ExtraSensory dataset. The resultant datasets are described in Table 1. Additionally, we show an example of one person's accelerometer data in Figure 1, showing at what timesteps they indicated they were *At a Bar*, one of our four labels.

## 4  EXPERIMENTS

We compare the performance of four machine learning algorithms using three preprocessing techniques for each of the four labels we extract from ExtraSensory. In all experiments, we evaluate

**Table 2: Balanced accuracy for down-sampling both train and test set. Each task corresponds to a unique binary classification dataset defined by the four self-reported labels described above.**

| Task | Method | | | |
|------|---------------------|----------|------------------------|------------------------|
|  | Logistic Regression | XGBoost | Support Vector Machine | Multi-layer Perceptron |
| At a Bar | 0.50 (0.08) | 0.74 (0.10) | 0.58 (0.06) | 0.61 (0.14) |
| At the Gym | 0.60 (0.10) | 0.76 (0.03) | 0.72 (0.07) | 0.72 (0.06) |
| At a Party | 0.75 (0.06) | 0.80 (0.04) | 0.77 (0.05) | 0.79 (0.04) |
| On the Bus | 0.55 (0.05) | 0.77 (0.02) | 0.75 (0.04) | 0.71 (0.04) |

**Table 3: Balanced accuracy for down-sampling training set and unmodified test set.**

| Task | Method | | | |
|------|---------------------|----------|------------------------|------------------------|
|  | Logistic Regression | XGBoost | Support Vector Machine | Multi-layer Perceptron |
| At a Bar | 0.50 (0.06) | 0.70 (0.06) | 0.57 (0.07) | 0.56 (0.06) |
| At the Gym | 0.64 (0.05) | 0.77 (0.03) | 0.72 (0.05) | 0.74 (0.03) |
| At a Party | 0.72 (0.05) | 0.81 (0.04) | 0.79 (0.05) | 0.81 (0.05) |
| On the Bus | 0.53 (0.03) | 0.79 (0.02) | 0.77 (0.02) | 0.75 (0.02) |

**Table 4: Balanced accuracy for SMOTE up-sampling training set and unmodified test set.**

| Task | Method | | | |
|------|---------------------|----------|------------------------|------------------------|
|  | Logistic Regression | XGBoost | Support Vector Machine | Multi-layer Perceptron |
| At a Bar | 0.58 (0.07) | 0.59 (0.08) | 0.59 (0.08) | 0.57 (0.05) |
| At the Gym | 0.69 (0.05) | 0.69 (0.04) | 0.70 (0.05) | 0.70 (0.04) |
| At a Party | 0.78 (0.05) | 0.77 (0.07) | 0.77 (0.06) | 0.72 (0.07) |
| On the Bus | 0.61 (0.04) | 0.57 (0.05) | 0.58 (0.05) | 0.53 (0.03) |

Balanced Accuracy, which weights accuracy according to the prevalence of each class. This is important as with unmodified testing set, standard accuracy will dramatically favor the majority class.

## 4.1 Compared methods and implementation

We compare Logistic Regression, XGBoost [5], a Support Vector Machine [6], and a Multi-layer Perceptron, all implemented in scikit-learn [20]. We use 10-fold cross validation, and report 95% confidence intervals with each reported statistic.

## 4.2 Results

*4.2.1 Downsampling training and testing data.* First, we downsample training and testing data to acquire one fully-balanced dataset per label, the results for which are shown in Table 2. This setting is highly unrealistic and serves as an upper bound for our next experiments; it assumes that at test time, any given person has a positive label 50% of the time. Correspondingly, the resultant accuracy is generally the highest we observe across our experiments. We notice that Attending a Party is the most predictable label and that the 95% confidence bounds overall are quite small. Still, in some cases predictions are very poor. For example, Logistic Regression fails to classify *At a Bar* or *On the Bus*.

*4.2.2 Downsampling training negatives.* Next, we try downsampling negative instances in the training sets for each label. This

way, we create a balanced training set and leave the testing set untouched and imbalanced. As shown in Table 3, we find similar performance as the fully-balanced case above across the board. For some classes, like *At a Bar* performance drops substantially.

*4.2.3 Upsampling training positives.* Finally, we overcome label imbalance by upsampling positive instances in only the training dataset using SMOTE. Our results are shown in Table 4. This approach augments the positive class by in-filling the positive instances in the feature space. While this may amplify signals found in small sets of positive instances, we find that performance slightly improves, even beyond the fully-balanced experiment above. This bolsters our intuition that these data are highly noisy and that synthetic instances surrounding real positive instances are also highly reasonable compared to the testing set. Once again, Logistic Regression appears to perform the worst, while XGBoost is highly successful.

## 5 ETHICAL CONSIDERATIONS

Detecting COVID-risky behaviors poses significant risk of disproportionately impacting historically-disadvantaged groups. This is because using cellphone data requires individual data access. Aggressive policing using these data has a long history of harming some groups more than others, so presenting already-biased institutions with more reasons to explain away privacy breaches

has potential to significantly increase harms. Given advances in privacy-preserving machine learning, we believe that there are benefits to be found at the *population* level. For instance, allowing public health officers to answer valuable questions about how effective their strategies are. Therefore, detecting COVID-risky behaviors could stand to improve public health, as long as it is not at the expense of vulnerable people. To this end, the individual data must be anonymized and aggregated to a population level before being made available to the surveilling institution.

Another consideration when detecting COVID-risky behavior is the inherent noise during prediction. Human Activity Recognition data can be quite noisy, which may lead trained models to produce false alarms. In such cases with untrustworthy models, the loss of privacy becomes more severe as no value was gained from the data. Further, if a model picked up on spurious correlations during training, its value is substantially decreased.

## 6 DISCUSSION AND CONCLUSION

We find that some COVID-risky behaviors are predictable given raw smartphone sensor data. If collectable at scale, data mining techniques for smartphones may add a tool to the public health official's toolbox. Further, by quantifying a population's behavior as an aggregate of individual activities, the feedback loop for public health interventions may be expedited. With faster feedback, a local health officer would be empowered to alter their messaging strategies rapidly, adding the adaptability crucial to early-management of communicable diseases. We suspect that similar findings are likely given different modalities, like smart watches and smart rings, and picking relevant activities to predict will be a valuable task for public health officials in the not-so-far future. However, there is no foreseeable harm-free application of these ideas. Successful use of mobile data for health must consider the health for *all*, especially those who stand to lose the most from aggressive surveillance.

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
