# OpenReview forum: "On Detecting COVID-Risky Behavior from Smartphones"
_ACM.org/SIGKDD/2022/Workshop/epiDAMIK — KDD 2022 Workshop epiDAMIK Oral_

### Official Review · Reviewer_jTak · 2022-06-19
**This paper conducts 4 binary classification tasks separately, using 4 of the widely used classification algorithms on a public smartphone sensor dataset. Though the contribution of this paper to epi domain is limited, this paper explores an interesting dimension of the usage of sensor data for detecting potential COVID-risky behavior.**

**Rating:** 3
**Confidence:** 3

**Review:**

This paper conducts 4 binary classification tasks separately, using 4 of the widely used classification algorithms on a public smartphone sensor dataset. Following are the strengths, weaknesses, and suggestions of the paper.

- Strengths

The paper is easy to read, and the results are clearly stated.

- Weaknesses

  - The authors' observation is that 'going to a bar', 'going to the gym', 'going to a party', and 'riding on the bus' are 'COVID-Risky behavior'. Based on this observation, they classify these four labels using a public dataset. Since this is the only connection of their method to COVID, a thorough literature review is needed to back up their observation. Otherwise, it's hard to find a connection to epi domain.
  - Their first experiment of 'downsampling both training and test dataset' is problematic. The test dataset should not be sampled.
  - The authors do not compare the performance to any other baseline methods.

- Suggestions

  - I suggest the authors make a solid connection that 'going to a bar', 'going to the gym', 'going to a party', and 'riding on the bus' is indeed a 'COVID-Risky behavior'. A survey on COVID patients' contact tracing showing that these were places where they got COVID may be enough.
  - I suggest authors remove the experiment and results that correspond to 'downsampling both training and test dataset.'
  - I suggest authors do a literature review on methods that mines data from the smartphone sensor data. Sensor data contain time-dependent information, which can be better captured using deep learning methods such as variants of RNNs or attention-based methods. This is an active research field, so do a survey on the methods, choose some of the methods as baseline methods, and compare the performance of the methods to the baselines.

---

### Official Review · Reviewer_dGrV · 2022-06-23
**The paper looks into an important aspect combining sensor data, ML and Healthcare. The work shows promising results and has multiple avenues in terms of future work**

**Rating:** 3
**Confidence:** 4

**Review:**


The paper is well written and discusses the idea of using sensory data from smartphones and wearables to understand epidemic risky behavior (COVID-risky in this case). Given the impact of the current pandemic, this work is relevant to both the workshop as well as the broader computational health community.

Pros.
1) The paper makes use of sensor data such as accelerometer data and therefore has a lesser impact on privacy.
2) The work aims to reduce the response time between public health messaging and learning about the impacts associated.
3) Make use of simpler models that can be further subjected to explainability checks given the nature of the problem (the current paper doesn’t look into it, but the authors use simple models thereby making such an extension possible)

Cons.
1) It would be interesting to show how this method is more effective or useful than normal contact tracing. A few of the previous works focused on privacy-preserving contact tracing (Shankar et al., 2020; Raskar et al., 2020).
2) The data sample only makes use of 60 people who provided data in various situations. It would be interesting to see the effects and reproducibility of the research on larger data sets.
3) The paper mentions how simple techniques like upsampling and downsampling have helped increase performance. There are recent studies like Goorbergh et al., (2022) that point out some common issues that arise with these methods. It would be interesting to know about the authors’ thoughts on the same.
4) Apart from self annotations done by the subject, where the data points are subjected to clustering or random sampling to see how different the various sensor values are between different activities - going to the gym vs going to a bar, etc. it would be interesting to see some quantification on how far/close these readings are in the input or latent space.
5) Although the paper mentions the importance of feedback loops based on public health messaging, I could not find the models taking in the context of public health instructions for the given data. That is, we don't know whether there was an instruction not to go to the bar or party. The model would be much more powerful if we can predict these population behaviors controlled by the public health messaging (In its current form, we can't be sure if this was a non-compliance or if there were no restrictions in the first place)

Overall I believe this is important work and would like to see the authors extend this work on larger datasets and by also combining with EN-related data as well as other modalities of data.

Refs.

https://arxiv.org/abs/2202.09101
https://arxiv.org/abs/2009.04991
https://arxiv.org/abs/2003.08567